# The feasibility of using ecological momentary assessment to understand urinary and fecal incontinence experiences in adults with spina bifida: A 30-day study

**Devon J. Hensel** [1,2,3]*, **Audrey I. Young**[3], **Konrad M. Szymanski**[4]

**1** Department of Pediatrics, Section of Adolescent Medicine, Indiana University School of Medicine, Indianapolis, Indiana, **2** Department of Sociology, Indiana University Purdue University-Indianapolis, Indianapolis, Indiana, **3** Department of Biology, DePauw University, Greencastle, Indiana, **4** Division of Pediatric Urology, Riley Hospital for Children at Indiana University Health, Indianapolis, Indiana

* djhensel@iu.edu

**Data Availability Statement:** All raw data files are available on the Open Science Framework (osf.io/gefqx).

## Abstract

In this paper, we evaluate the feasibility of using ecological momentary assessment (EMA) to understand urinary (UI) and fecal (FI) incontinence in adults with spina bifida (SB). As part of a larger 30-day prospective study to understand the incontinence in adults with SB (N = 89), participants completed end-of-day EMA diaries assessing the frequency and context of UI and FI. We used these data to assess the method feasibility across six dimensions: (a) compliance, or data entry which is consistent with study protocol and substantially complete; (b) reactivity, or behavior change attributed to study participation; (c) participant acceptability, or convenience and ease of method beneficial to compliance; (d) data capture, or the volume of incontinence behaviors collected; (e) the accuracy of incontinence reports; and f) participant-provided feedback for future studies. Participants were highly compliant with diary entry protocol and schedule: submitting 95.7% (2576/2700) of the expected total daily entries. The average completion time was two minutes. Neither the total number of submissions nor the completion time varied by demographic characteristics or health history. A sufficient volume of incontinence and affective outcomes were captured, with small downtrends in reporting of UI and affect over time. Exit survey recall was highly correlated with diary reports. Participants found the methodology to be acceptable, reported their experiences honestly, enjoyed and felt comfortable participating in the study and would engage in similar study in the future. Accurate information about the daily context of UI and FI is a key factor in the success of intervention or education programs relying on this information. Our findings demonstrate that EMA is a feasible way to describe UI and FI in adults with SB.

## Introduction

Spina bifida (SB) is a congenital anomaly in which the neural tube does not fully close during embryonic development, resulting in permanent damage to the spinal cord [1]. Depending on

**Funding:** This project was supported by grant R21 HD053231-01A1 to DJH and KS from the National Institute of Diabetes Digestive and Kidney Disorders (NIDDK: https://www.niddk.nih.gov/). The contents of this paper are solely the responsibility of the authors and do not necessarily reflect the official views of the NIDDK. The funder had no role in study design, data collection and analysis, decision to publish, or preparation of the manuscript.

**Competing interests:** Dr. Hensel is a paid research consultant with For Goodness Sake, LLC.

**Abbreviations:** EMA, ecological momentary assessment; SB, Spina Bifida; UI, urinary incontinence; FI, fecal incontinence.

defect severity, people with SB can experience several different chronic health complications, including urinary (UI) or fecal (FI) incontinence [2]. Three quarters of adults with SB report UI, half experience either FI or many live with both UI and FI [3]. Incontinence is broadly recognized to decrease mental health (e.g. increased depression or anxiety) and social outcomes (e.g. interference with employment, daily activities and relationships) [4–6], and our own research additionally suggests that specific dimensions of UI and FI (e.g. quantity vs. frequency vs. amount) are differentially linked to lower quality of life [3, 7–10]. The high prevalence and impact of UI and FI in the SB population highlight the need for data collection methods that can capture daily incontinence experiences with fidelity to the ways in which people actually live them [11].

Most SB research assesses incontinence by way of retrospective questionnaires that require individuals to self-recall and summarize incontinence information after what is typically a time delay (e.g., "On average, how many times a day did you experience urinary incontinence in the past week?" or "In general, how bothersome was your fecal incontinence over the past month?") [12, 13]. This approach is common because it is difficulty for a clinician or researcher to directly observe incontinence in a person's daily life. At the same time, however, these approaches present challenges. Sensitive and/or stigmatized behaviors, like incontinence, can be underreported out of participant concern for privacy or judgement from others [14]. Longer recall periods may decrease the accuracy with which a participant can recall or describe incontinence experiences [15]. Finally, summarizing multiple sources of information into a single data point (for example, aggregating 30 individual days into a single "month" focused question) removes the ability to understand how day-to-day changes in different aspects of incontinence, like frequency, bother or interference, could have different impact on a person's well-being [16–18]. All three of these issues–underreport, recall accuracy and lack of event-specific detail–can degrade the reliability and the validity of the information collected.

Recent studies have shown that ecological momentary assessment (EMA) is a promising method to maximize the "real-life" generalizability of self-reported health behavior data. EMA refers to a broad class of prospective–or forward looking–data collection approaches intended to capture snapshots of people's daily lives through brief-but-repeated assessments in their "natural" environments [19]. "Ecological" refers to collecting data as feasibly close in time and condition to when people themselves experienced them [20, 21], while "momentary" refers to people's repeated reporting of events or experiences at the current moment or very recently in time [22]. Data are typically collected using the functional capabilities of web-enabled electronic devices, and participants provide data on a study-specific time schedules (for example, once/day or following a specific health event; for an excellent review, see Hufford et al., 2007 [23]) usually over weeks or months [24, 25]. Used in the context of incontinence, EMA is ideal for amassing enough data points to be able understand of how a participant's personal or social context differs when UI or FI does or does not occur, as well as how the relationship between context and UI or FI may change over time [26]. Such data are important for optimizing the chance that interventions based on this knowledge will be effective when employed in people's daily lives [27].

There are data quality, programming and data security advantages to using EMA (for excellent reviews, see Hensel, 2014 [28] or Hufford and Shiffman, 2002 [26]). From a *data quality* perspective, EMA typically produces higher reporting rates, lower levels of missing data, more robust internal validity and lower behavior reactivity (e.g. changing what is reported as a result of participating in the study)compared to other collection approaches [20, 22]. From a *programming* perspective, EMAs can be designed to be delivered across many different types of web-enabled devices (e.g., smartwatches, cell phone, tablets) [29]. There is incredible flexibility in question structure (e.g. fixed choice vs. free text, single choice vs. "check all that apply,"

check box or sliding "rating" type scales, etc.), in assessment frequency (e.g. once per day/week vs. multiple times/day) and in assessment pattern (e.g., at the same time, only when an event happens, or at random intervals) [28]. Multiple language versions of surveys can be offered [30]. All interactions a participant has with the electronic system can be time-stamped, allowing real-time description of compliance (e.g., number of diaries started, completed, or submitted), and/or calculation of average completion time for specific questions, or for the entire diary [31]. In some instances, a diary could be paired with other electronic 'add-ons' to enhance or supplement existing data (e.g., 'apps,' video-sharing, chatting) [28]. From a *data security* perspective, EMA can strengthen the security of sensitive or stigmatizing information by allowing information to be vacated from the device immediately upon data entry for storage on a remote server, thereby increasing anonymity and confidentiality [16]. Participants can additionally be encouraged to use the security features (e.g., passcode, Face ID, or fingerprint ID) on their device.

Perhaps most importantly, the electronic structure of EMA additionally confers *participant-centered* advantages for populations of disabled persons who may otherwise be excluded from clinical research [32]. Mobility challenges create research participation barriers for individuals with SB, either because travel to non-clinical recruitment sites is difficult, or the specialty clinic at which they receive care is not a recruitment site [33]. The electronic nature of EMA allows data collection to happen quite literally "anywhere and anytime" [34, 35], removing nearly all the constraints of specific times or physical spaces in which research must be conducted [36]. Web-enabled devices, such as cell phones and tablets, are the most commonly owned device by people with disabilities [37], including those with SB [38, 39]. Adults and young people in this population use their web-enabled technology daily for various purposes, such as health care, work, education and social interaction [40–43]. Navigating these different tasks means that individuals with SB are already "self-trained" on most, if not all, of the physical skills needed for study research participation (e.g., clicking on links, opening documents, entering/changing text, using fingers to swipe pages or mark boxes or submitting forms) [44]. Engaging technology with which individuals are already actively using permits easy introduction of EMA in a manner that is minimally invasive and minimally disruptive to their daily life [45]. Participants are able to choose where and when they complete data entry and submission, allowing them greater flexibility in participating in a study and potentially increasing their comfort in disclosing sensitive information like incontinence [28].

EMA has been utilized to examine a variety of clinical health outcomes in populations both with and without disabilities, including chronic pain [25], chronic illness in association with disability [46], hearing aid use among individuals with hearing impairment [47], neuropathic pain among individuals with spinal cord injury [24], daily functioning in individuals with schizophrenia [48] and daily affect following traumatic brain injury [49]. EMA has also been engaged to examine urinary symptoms among prostate cancer patients [50] and bowel incontinence among individuals with different conditions [51, 52]. EMA has not yet been used in clinical research among individuals with SB.

In this paper, we evaluate using daily EMAs over 30 days as a means of prospectively understanding experiences of UI and FI among adults with SB. Using a feasibility framework established by the first author's work [16, 17, 53, 54], we assess (a) compliance, or data entry which is consistent with study protocol and substantially complete; (b) reactivity, or behavior change attributed to study participation; (c) participant acceptability, or convenience and ease of method beneficial to compliance. We also assess (d) data capture, or the volume of incontinence behaviors collected and (e) the accuracy of incontinence reports. Finally, we review participant-provided feedback on ways to improve future studies.

## Methods

### Study design and data collection

Data were collected as part of a larger 30-day study prospectively examining the daily prevalence and context of UI and FI in adults with SB. In this larger study, data collection was organized around three components: 1) a self-administered enrollment module, which allowed participants to provide informed consent and HIPPA authorization, as well as assessed demographics, quality of life, and past behavioral and social experiences with incontinence; 2) once daily EMA diaries monitoring affect, health, frequency, volume and management of UI and/or FI reported, as well as daily activities; 3) a self-administered exit survey which assessed participant experiences in the study, as well as gathered retrospective reports of incontinence variables reported in the EMA diaries. All data collection was automated, meaning that the completion of one data source intuitively triggered the delivery of the next data source. For example, submission of the enrollment module prompted the distribution of the Day 1 EMA, the completion of the Day 1 EMA signaled the delivery of the Day 2 EMA, and so on, and the completion of the last EMA triggered the delivery of the exit survey. The larger project was approved by the Institutional Review Board of Indiana University (#1907916729).

All data collection was conducted electronically via a secure, cloud-based survey tool called Qualtrics [55]. We chose this program for several reasons. The program is accessible from multiple devices (e.g. a phone, tablet or laptop/desktop computer) via any internet connection and any web browser. In addition, Qualtrics offers intuitive programming of different survey types (e.g. cross-sectional vs. longitudinal) and question types (e.g. scales, open-ended, single- and multiple-choice, and electronic signature). The program additionally does not require that participants have additional tools (e.g., stylus) or specialized software (e.g., Adobe Acrobat) [56] to provide data. These two attributes–flexible accessibility and question type–means generally increased ease-of-access for unique populations needs. For example, participants who have multiple devices (e.g. they use a phone and tablet interchangeably) as well as participants who only have one device (e.g. a laptop) are equally able to complete data collection regardless of any personal limitations. Qualtrics also offers a flexible project management interface, allowing researchers to quickly be able to see and organize multiple data arms for the same study. After data collection, researchers are able to engage several different automated download options to house their data in different programs (e.g. Excel or statistical software).

### Participants

Due in part to both COVID-19 clinical restrictions during the recruitment period (e.g. no research personnel were allowed in clinical waiting rooms from which participants would have otherwise been recruited) and to the common physical limitations of many individuals with SB (e.g. mobility challenges make in person recruitment less possible), we engaged a combination of electronic convenience and snowball sampling methods as means of reaching potential participants. Short study announcements were placed on the social media platforms (e.g., Twitter, Facebook, Instagram) of local, national and international SB (e.g., the national Spina Bifida Association, Spina Bifida Indiana) organizations contacted directly by the investigators. In addition, study fliers were mailed to local SB patients receiving care at our center who had agreed to receive study-related information. All announcements and fliers contained a QR code that, when scanned, took interested individuals to a study landing page that provided more detailed study information and the contact information of study staff.

Study eligibility requirements included being 18 years or older, having SB, having any UI and/or FI in the past four weeks, having English language literacy and having normal to mildly

impaired cognitive development. Specific device ownership was not a requirement, and we built in study funds to buy a web-enabled phone for any participant who either had no way of accessing the internet for data collection, or who preferred not to use their own device. All participants opted to use devices already in their possession. All potential participants made an appointment for a virtual information and screening session with a member of the study staff. During the appointment, staff reviewed the study purpose, data collection requirements, and compensation schedule with individuals, along with language literacy and developmental delay. Those who passed screening were emailed a link to the enrollment module where they completed informed consent, HIPAA authorization and provided background information. Within this module, they also chose a delivery method–short message service (SMS; e.g., a text message) or email–for the remaining data sources. About 80% of participants opted for the SMS option, and no one asked to change their first chosen method.

## EMA diaries

During the enrollment appointment, study staff provided an in-depth orientation to the EMA diary entry protocol with all participants. During this time, they also reviewed the policies on missing entries and tips for troubleshooting procedures for equipment or connectivity issues. Beginning on the day after enrollment completion, participants received an SMS or an email at 7pm in their local time zone that linked them to that day's EMA diary. While some EMA researchers suggest prompting diary completion at random intervals each day [20], other suggests that data collection at consistent times each day is equally valid [57], and may increase the convenience of, and compliance to, the data collection procedure for participants.

Once participants clicked on the diary link, they arrived on the landing page of the Qualtrics survey (Fig 1, Panel A) associated with that specific EMA diary day in their study progression. Individuals completed a specific sequences of questions that assessed their experiences since their last entry, including their affect (example item, Fig 1, Panel B), UI or FI frequency (example items, Fig 1, Panels C and D) and daily activities. If any incontinence was reported, they answered contingency items about amount, management, activities they avoided and

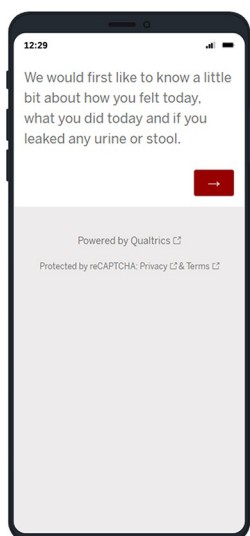 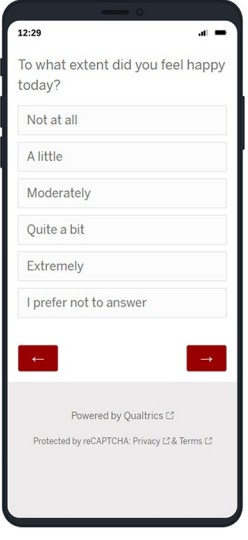 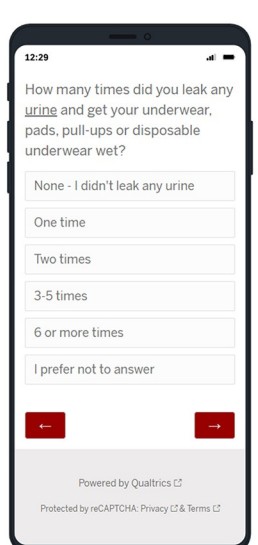 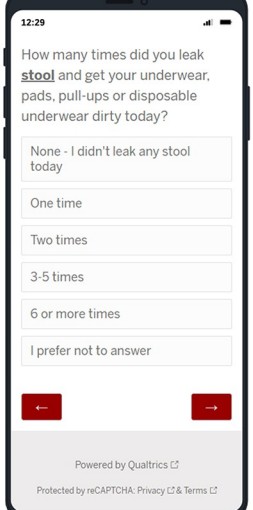

| Panel A. Home screen | Panel B. Affect item | Panel C. Urinary incontinence frequency item | Panel D. Fecal incontinence frequency item |

**Fig 1. Example EMA diary screens.**

feelings about the incontinence. If *no* incontinence was reported, participants reported on their worry about any incontinence happening and any activities they avoided around that worry. Diaries were formatted to display one question on the screen at a time, requiring participants to select from a drop down list, or check appropriate boxes. Navigation from question to question was facilitated by the use of a "previous" and "next" button on each screen; however, for most questions, participants could not advance until the current question had been answered. For some sensitive questions, a "Prefer Not to Answer" option was available. These features were built into the diary program to reduce data errors and omissions.

Participants could access that day's EMA diary at any time for a window of four hours, either completing it in one sitting, or completing it in chunks, as data were saved as the participant entered it. No information resided on the participant's device. Completion reminders were sent via the original delivery method every 15 minutes during the four-hour window until either the EMA diary was complete, or the time window closed. Participants who accessed the diary after the window closed were unable to get into (or back into) that specific entry, and anything not completed was considered missing. Study staff monitored expected completion daily from actively enrolled participants and contacted anyone who missed more than two consecutive entries. Such monitoring was important given that subsequent data collection was attached to the completion of prior entries. In the event that a participant reported that they did not receive an entry (e.g. they were out of cellular tower signal range, or their phone malfunctioned), or they had a reasonable excuse as to why they could not complete the entry within the window (e.g. a family emergency or they were unexpectedly out of town) were permitted to complete their entry either outside the window on that same day, or they were given a replacement entry for the next day. We worked to accommodate participants where at all possible.

## Participant compensation

Compensation amounts were structured using both standards of prior work conducted by the first author and acknowledgment of participant's sharing of personal and potentially sensitive health information. Participants received $15.00/each for the completion of the enrollment and exit modules, and $0.50 for the submission of each EMA diary. To incentivize high completion of the diary arm, we added a $10.00 bonus for any participant who provided 90% or more of all expected diaries. In addition, individuals who completed all study arms received $30.00 to offset the cost of text messaging and web access during the study. In total, it was possible to earn $85.00 in this 30 day study. Participants were paid via gift card of their choosing at the end of the study. All non-completers were paid for whatever portions they completed.

## Measures

For the purposes of feasibility examination in different ways, we analyzed different variables collected in the EMA diaries. Incontinence variables included: *UI frequency* and *FI frequency* (both: no times, one time, two times, 3–5 times or six or more times). Emotion variables included: *daily affect* (PANAS short form; positive mood [Cronbach's alpha: 0.987] and negative mood [Cronbach's alpha: 0.965], both 3-item using 5-point Likert type items [strongly disagree to strongly agree]; e.g. "I felt happy" or "I felt sad") and health (single 4-point item: poor to excellent) [58, 59] and on days when UI or FI was reported, *incontinece negativity* (4-item using 5-point Likert type items [strongly disagree to strongly agree]; e.g. "I felt [anxious, bothered, stressed, frustrated] about having [urinary or fecal] incontinence today" [Cronbach's alpha: 0.968 [UI] and 0.976 [FI]). We finally examined two daily behaviors (both no/yes): *watching TV/movie* and *eating*.

## Analyses

We used descriptive statistics, including percentages, to examine participant *compliance* (e.g. the extent to which participants complete the study and provide data that aligns with study protocols), participant EMA diary *completion* (e.g., the daily, weekly and study level degree to which diaries are submitted), participant *acceptability* (e.g., perceptions of method ease and convenience, question length, interaction with study team and willingness to participate in future research: each on a 5-point strongly disagree to strongly agree Likert scale). Open ended, free text responses on the exit survey were used to additionally inform acceptability and to describe *suggestions for future studies*. We also engaged descriptive statistics to evaluate *data capture*. Finally, we used Spearman's rank correlation–a nonparametric substitute for Pearson's R when count data (e.g. the number of UI or FI events) are used [60]–and descriptive statistics to assess *data accuracy*. Effect size interpretation guidelines for Spearman's rank correlation were taken from Schober et al. [61].

We examined *reactivity* (e.g., the extent to which participant data change as a function of being in the study) in three domains: participation, behavioral and emotional. Participation reactivity was operationalized using two indicators of study engagement, including EMA completion time (minutes) and total number of EMAs completed. We evaluated behavioral reactivity using a measure of daily UI and FI (both: no/yes), as well as a pair of daily activities, including watched TV or a movie and ate a meal (both no/yes). These chosen measures aided in comparing if and/or how reactivity may differ between incontinence and more routine activities. Emotional reactivity was characterized using positive and negative mood, as well as UI or FI incontinence negativity.

In all models, we used random intercept mixed effects conditional linear (for completion time, mood and incontinence negativity), Poisson (for number of EMAs contributed) or binary logistic (for UI, FI and daily activities) with time (in days) as the sole predictor of each outcome. A significant value indicated a change in reporting as a function of time. For any significant coefficient, we converted the estimate to a percent via the first derivative for better interpretation. Random intercepts allowed estimates to be adjusted for the multiple diaries provided by participants [62, 63]. All data analysis was performed in Stata 17.0 (critical p = 0.05) [64]. Data were de-identified prior to analysis [65].

## Results

### Participants and retention

A total of 90 individuals enrolled in the study (May 2020-November 2020), and 97.8% of these (N = 88) completed all three expected arms of the project. Two individuals completed enrollment, but were dropped from the study after completing, respectively, none of the EMAs and only the first nine EMAs. Both of these participants were contacted multiple times by the study coordinator via text message, email and voice call to troubleshoot any aspects of EMA completion that could have been challenging. Neither individual responded to any of the attempted contacts, and after ten days were removed from the study per study protocol.

Table 1 provides the demographic and background characteristics of the completed participants. Participants were primarily female identified (71.1%), White (86.7%) and had an average age of 35 (range: 19–72). About seven in ten lived independently and had a median highest education level of a college degree. All participants had residence in the United States, Canada, South Africa and the United Kingdom, and half were working part or full time (51.1%). About a third were single and not dating. Nearly all (88.0%) were heterosexual. Two-thirds of participants self-classified as community ambulators. While all participants reported either UI or FI

**Table 1. Demographic characteristics of completed participants (N = 88).**

|  | Mean (SD); Median (IQR) or % |
|---|---|
| Age | 35.7 (11.7); 34.0 (26.5–41.0) |
| Any past four-week urinary incontinence |  |
| No | 6.7 |
| Yes | 93.3 |
| Any past four-week fecal incontinence |  |
| No | 20.0 |
| Yes | 80.0 |
| Birth-assigned sex |  |
| Female | 71.1 |
| Male | 28.9 |
| Race |  |
| White | 86.7 |
| African American | 7.8 |
| American Indian or Alaska Native | 0.0 |
| Asian | 0.0 |
| Native Hawaiian or Pacific Islander | 1.1 |
| More than one race | 2.2 |
| Other | 3.3 |
| Sexual orientation |  |
| Heterosexual | 88.0 |
| Sexual minority | 12.0 |
| Gay or lesbian | 2.2 |
| Bisexual | 6.5 |
| Asexual | 2.1 |
| Something else | 1.1 |
| Hispanic/Latino (yes) |  |
| No | 93.3 |
| Yes | 6.7 |
| Living situation |  |
| Independently (on own, with roommate(s) or partner) | 69.6 |
| Dependently | 30.4 |
| Yearly household income |  |
| <20,000 | 21.1 |
| 20,000–39,999 | 18.9 |
| 40,000–59,000 | 10.0 |
| 60,000–79,999 | 8.9 |
| 80,000–100,000 | 6.7 |
| >100,000 | 12.2 |
| Not sure | 10 |
| Highest level of education |  |
| HS degree or equivalent (e.g. GED) | 17.8 |
| Some college but no degree | 20 |
| Associate degree or Technical/Professional degree | 7.8 |
| Bachelor's degree | 34.4 |
| Graduate degree | 20.0 |
| Work Status |  |
| Working full-time | 31.1 |

*(Continued)*

**Table 1.** (Continued)

| | Mean (SD); Median (IQR) or % |
|---|---|
| Working part-time | 20.0 |
| Unemployed and looking for work | 8.9 |
| Unemployed and not looking for work | 12.2 |
| On disability | 25.6 |
| Relationship status | |
| Single and not dating | 36.7 |
| Single and dating, but not in a relationship | 10 |
| In a relationship, but not living together | 13.3 |
| Living with a partner, but not married | 8.9 |
| Married | 27.8 |
| Divorced or separated | 3.3 |
| Country of Residence | |
| United States | 95.6 |
| Canada | 2.2 |
| South Africa | 1.1 |
| United Kingdom of Great Britain and Northern Ireland | 1.1 |
| How you get around | |
| Non-community ambulator | |
| I move only in my wheelchair | 31.1 |
| I walk only during therapy, wheelchair the rest of the time | 2.2 |
| I walk inside, always use a wheelchair outside | 2.2 |
| I walk inside and outside, only use a wheelchair for long trips | 21.1 |
| Other | 15.6 |
| Community ambulator | 27.8 |
| I walk inside and outside, without any help | 27.8 |
| Have a shunt | |
| No | 46.7 |
| Yes | 53.3 |
| How do you empty your bladder | |
| Urinate in toilet w/o pushing | |
| No | 87.8 |
| Yes | 12.2 |
| Push to get urine out | |
| No | 84.4 |
| Yes | 15.6 |
| Urine leaks from urethra/below | |
| No | 90 |
| Yes | 10 |
| Catheterize | |
| No | 25.6 |
| Yes | 74.4 |
| Wear stoma/bag | |
| No | 97.8 |
| Yes | 2.2 |
| Other | |
| How do you empty bowels (stool) | |
| Catheterize to flush water in my stoma made by a surgeon | 17.8 |

(*Continued*)

**Table 1.** (Continued)

|  | Mean (SD); Median (IQR) or % |
|---|---|
| Wear a bag on my abdomen that collects my stool | 4.4 |
| Other | 73.3 |
| Prefer not to answer | 4.4 |
| Last 4 weeks, how long you stayed dry between urine accidents | |
| Never dry | 10.0 |
| Less than 4 hours | 28.9 |
| 4 hours or longer | 54.4 |
| Prefer not to answer | 6.7 |
| Last 4 weeks, how much urine you usually leaked | |
| A little bit (a drop) | 23.3 |
| Medium amount | 53.3 |
| A lot | 16.7 |
| Prefer not to answer | 6.7 |
| Last 4 weeks, how you managed cleaning leaked urine | |
| I took care of it on my own | 83.3 |
| I needed a little help | 2.2 |
| I needed a lot of help | 7.8 |
| Prefer not to answer | 6.7 |
| Last 4 weeks, how long you stayed clean between stool accidents | |
| Never clean | 2.2 |
| Less than 1 week | 28.9 |
| 1 week or longer | 48.9 |
| Prefer not to answer | 20 |
| Last 4 weeks, how much stool you usually leaked | |
| A little, a smear | 38.9 |
| Medium amount | 30 |
| A lot | 11.1 |
| Prefer not to answer | 20 |
| Last 4 weeks, how you managed cleaning leaked stool | |
| I took care of it on my own | 64.4 |
| I needed a little help | 7.8 |
| I needed a lot of help | 7.8 |
| Prefer not to answer | 20 |

in the past 4 weeks, 93.3% reported UI, 80.0% reported FI and 73.3% reported both. The median length of time between UI events was four hours or longer, and a week or longer between FI events. During incontinence accidents, participants most commonly reported leaking a "medium amount" of urine (53.3%) and a "little bit" of stool (38.8%). Nearly all participants managed clean up on their own (UI: 83.3%; FI: 64.4%).

## Compliance

We analyzed compliance analysis at three levels. First, participants were asked to complete one EMA entries per day, resulting in seven expected submissions *per week* during the first three weeks of the study (e.g., Week 1: Days 1–7; Week 2: Days 8–14; Week 3: Days 15–21) and nine entries in the last week (e.g. Week 4; Days 22–30). Second, each *participant* was expected to complete a total of 30 entries by the end of the study. Third, at the *study level*, the 90 completed participants were expected to contribute 2700 (90 x 30) entries by the end of the study [66].

Compliance was excellent at all three levels. At the *weekly* level, participants submitted an overall mean of 6.64 (SD = 0.79; Median = 7) of the seven (94.6%) expected weekly diaries during the first three weeks of the study, and a mean of 8.53 (SD = 1.01; Median = 9) of the nine expected entries (94.7%) during the fourth week. At the *participant* level, the majority (84.4%: 76/90) submitted 28 or more of the total expected 30 diaries at the end of the study. At the *study* level, participants submitted 95.7% (2576/2700) of the total diaries expected.

A preliminary examination of the distribution of completion time at the weekly level (Mean: 7.2; Median: 7.0; Mode: 7.0) and the participant level (Mean: 29.3; Median: 30.0; Mode: 30.0) suggested that the variable was normally distributed. Therefore, values in Tables 2 and 3 are represented by the mean. Table 2 provides a detailed illustration of EMA as a function of a specific day in the study. On each day, a total 90 entries–one per participant–were expected. As shown in the third column "Number of EMAs submitted") the total number of EMA submissions on each day ranged between 81 (90%) and 90 (100%) (daily median: 86; data not shown). As shown in Table 3, we examined EMA submission at both the participant and weekly levels as a function of demographic and incontinence history characteristics. There were no significant differences observed in the average number of EMAs submitted–at either level–by any of the chosen factors.

## Completion time

We operationalized completion time as the number of minutes elapsed from when a participant accessed the first question (Fig 1, Panel A) to when they submitted the diary by providing an answer to the last question. A preliminary examination of the distribution at the and weekly level (Mean: 35.4; Median: 2.0; Mode: 2.0) and participant level (Mean: 36.1; Median: 2.0; Mode: 2.0) suggested that the variable was positively skewed. Therefore, values in Tables 2 and 3 are represented by the median. The overall median time to completion was two minutes and 75% of all responses were submitted within three minutes.

Table 2 provides a detailed illustration of EMA as a function of a specific day in the study. As shown in the second column ("Completion Time") the median completion time on each day ranged between 1.8 minutes and 3.3 minutes. As shown in Table 3, we examined completion time at both the participant and weekly levels as a function of demographic and incontinence history characteristics. There were no significant differences observed in the completion time–at either level–by any of the chosen factors.

## Reactivity

We evaluated reactivity in three domains–participation, behavioral and emotional–by using time (in days) as a predictor of each outcome. The results of these models are shown at the bottom of Table 3 ("Time Effect"). Time did not have a significant effect on the participation measures, meaning that variability in both EMA submission and completion time were not linked to the passage of study time. Of all behavioral measures, time significantly predicted number of reported UI events and number of eating events. UI reports *decreased* by 2.1% between the first and last day of the study, and eating events increased by 2.9% in that same time frame. Finally, of all emotional measures positive mood was significantly associated with time: the median reported level decreased by 1.4% over the study.

## Acceptability

Table 4 displays participant acceptability ratings. Individuals highly rated their overall study experience: 90.9% agreed or strongly agreed (median: 4) that they "enjoyed participating in the study" and 88.5% agreed or strongly agreed that payment for their time was fair. Study team

**Table 2. EMA submission numbers and completion time by demographic and incontinence history–Participant level and weekly level.**

| Demographic and Incontinence History | Participant Level | | | | Weekly Level | | | |
|---|---|---|---|---|---|---|---|---|
| | Number of completed diaries[1] | p-value | Time to complete diaries (minutes) | p-value | Number of completed diaries[2] | p-value | Time to complete diaries (minutes) | p-value |
| | Mean (SD)[3] | | Median[3] | | Mean (SD)[3] | | Median[3] | |
| Gender Identity | | 0.251 | | 0.221 | | 0.216 | | 0.548 |
| Male | 29.54 (0.81) | | 2.08 | | 7.38 (0.90) | | 2.05 | |
| Female | 28.16 (4.08) | | 2.11 | | 7.09 (1.20) | | 2.10 | |
| Other | 30.00 (0.89) | | 1.86 | | 7.5 (1.0) | | 1.84 | |
| Sexual Orientation | | 0.953 | | 0.294 | | 0.317 | | 0.954 |
| Heterosexual | 28.57 (3.55) | | 2.04 | | 7.15 (1.16) | | 2.37 | |
| Sexual minority | 28.64 (3.23) | | 2.30 | | 7.14 (1.12) | | 2.02 | |
| Any past four-week urinary incontinence | | 0.222 | | 0.053 | | 0.526 | | 0.834 |
| No | 29.33 (1.21) | | 2.55 | | 7.29 (1.12) | | 1.59 | |
| Yes | 28.32 (3.60) | | 3.78 | | 7.17 (1.13) | | 2.11 | |
| Any past four-week fecal incontinence | | 0.213 | | 0.401 | | 0.612 | | 0.493 |
| No | 27.06 (6.14) | | 5.77 | | 6.99 (1.32) | | 1.91 | |
| Yes | 28.96 (2.35) | | 2.58 | | 7.22 (1.08) | | 2.10 | |
| Race (White) | | 0.259 | | **0.007** | | 0.158 | | 0.715 |
| No | 29.25 (1.60) | | 9.15 | | 7.31 (0.97) | | 2.28 | |
| Yes | 28.47 (3.69) | | 1.69 | | 7/16 (1.14) | | 2.05 | |
| Hispanic/Latino | | 0.949 | | 0.583 | | 0.2 | | 0.978 |
| No | 28,57 (3.51) | | 2.81 | | 7.18 (1.13) | | 2.06 | |
| Yes | 28.67 (2.10) | | 3.02 | | 7.16 (1.01) | | 2.07 | |
| Living situation | | 0.676 | | 0.669 | | 0.959 | | 0.95 |
| Dependently | 28.82 (2.76) | | 2.05 | | 2.02 (3.17) | | 7.19 | |
| Independently (on own, with roommate(s) or partner) | 28.48 (3.76) | | 2.09 | | 2.10 (4.70) | | 7.17 | |
| Community ambulator | | 0.976 | | 0.7 | | 0.884 | | 0.857 |
| No | 28.58 (3.95) | | 2.06 | | 2.10 (2.81) | | 7.19 | |
| Yes | 28.56 (1.73) | | 2.07 | | 2.03 (3.28) | | 7.14 | |
| Have a shunt | | 0.356 | | 0.284 | | 0.397 | | 0.795 |
| No | 28.93 (2.01) | | 2.05 | | 2.05 (1.59) | | 7.22 | |
| Yes | 28.27 (4.04) | | 2.06 | | 2.06 (2.91) | | 7.14 | |

[1]The expected number of entries at the participant level (e.g. per each participant) was 30.

[2]The 4th week has nine expected entries, making the overall weekly average slightly larger than seven.

[3]The mean was used to represent number of submissions and the median to represent completion time based on distribution shape. Additional data are provided in the manuscript.

support was also highly an important aspects to participants. A majority strongly agreed that they were both "able to communicate with the study staff when needed" and "felt supported by the study team throughout the study." Open-ended free text data support these themes; for example, one participant reported that they "liked being able to express [their] feelings regarding bladder and bowel control" and another suggested that it helped them "track [their] bowel issue more consistently."

We additionally evaluated the technical aspects of the study that were important to supporting this population's participation. Seven in ten strongly agreed that it was "easy to take the

**Table 3. Participation, behavior and emotional reactivity–EMA level.**

| Day number | Participation Reactivity | | Behavior Reactivity | | | | Emotional Reactivity | | | |
|---|---|---|---|---|---|---|---|---|---|---|
| | Completion time | Number of diaries completed | Number of UI events | Number of FI Events | Number of Watched TV/ Movie Events | Number of Eating Events | Positive Mood | Negative Mood | Negativity with UI | Negativity with FI |
| | Median | Sum | Sum | | | | Median | | | |
| 1 | 3.29 | 90 | 59 | 23 | 62 | 76 | 7 | 4 | 8 | 12 |
| 2 | 2.75 | 90 | 54 | 21 | 62 | 71 | 7 | 4 | 7.5 | 12 |
| 3 | 2.59 | 90 | 53 | 20 | 57 | 71 | 6 | 3 | 9 | 11.5 |
| 4 | 2.52 | 90 | 52 | 14 | 58 | 77 | 6 | 3 | 8 | 11.5 |
| 5 | 2.20 | 88 | 50 | 16 | 54 | 76 | 6 | 3 | 7 | 8 |
| 6 | 2.43 | 89 | 49 | 21 | 60 | 74 | 6 | 3 | 7 | 12 |
| 7 | 2.17 | 86 | 51 | 15 | 54 | 68 | 6 | 3 | 7 | 7 |
| 8 | 2.34 | 85 | 42 | 17 | 55 | 71 | 6 | 3 | 6 | 9 |
| 9 | 2.03 | 83 | 47 | 17 | 63 | 70 | 6 | 3 | 7 | 13 |
| 10 | 2.11 | 88 | 51 | 19 | 60 | 77 | 6 | 3 | 7 | 10 |
| 11 | 2.05 | 88 | 45 | 21 | 61 | 77 | 6 | 3 | 8 | 8 |
| 12 | 2.12 | 85 | 42 | 18 | 63 | 77 | 6 | 3 | 8 | 8.5 |
| 13 | 2.30 | 84 | 52 | 18 | 62 | 75 | 6 | 4 | 8 | 12 |
| 14 | 2.01 | 84 | 44 | 22 | 62 | 72 | 6 | 4 | 7.5 | 13 |
| 15 | 2.00 | 82 | 45 | 13 | 58 | 74 | 6 | 4 | 7 | 8 |
| 16 | 1.78 | 88 | 47 | 13 | 67 | 80 | 6 | 3 | 8 | 9 |
| 17 | 2.12 | 86 | 46 | 19 | 63 | 82 | 6 | 3 | 8 | 12 |
| 18 | 1.88 | 85 | 43 | 10 | 59 | 78 | 6 | 3 | 8 | 9.5 |
| 19 | 2.03 | 85 | 43 | 13 | 58 | 77 | 6 | 3 | 8 | 10 |
| 20 | 2.06 | 86 | 46 | 13 | 65 | 78 | 6 | 3 | 7.5 | 8 |
| 21 | 1.82 | 86 | 45 | 15 | 65 | 81 | 6 | 3 | 8 | 11 |
| 22 | 1.93 | 86 | 41 | 17 | 61 | 78 | 6 | 3 | 8 | 8 |
| 23 | 1.82 | 83 | 48 | 16 | 65 | 75 | 6 | 3 | 7 | 10.5 |
| 24 | 1.97 | 81 | 41 | 16 | 57 | 73 | 6 | 3 | 8 | 8 |
| 25 | 2.03 | 81 | 43 | 18 | 64 | 76 | 6 | 3 | 8 | 10.5 |
| 26 | 2.0 | 86 | 46 | 23 | 59 | 80 | 6 | 3 | 8 | 11 |
| 27 | 1.8 | 86 | 45 | 18 | 59 | 79 | 6 | 3 | 8 | 8 |
| 28 | 1.8 | 83 | 44 | 16 | 65 | 75 | 6 | 3 | 8 | 8 |
| 29 | 1.82 | 85 | 46 | 14 | 64 | 78 | 6 | 3 | 6.5 | 8 |
| 30 | 1.72 | 87 | 46 | 12 | 64 | 78 | 6 | 3 | 8 | 8 |
| Overall Total or Median | 2.03 | 2576 | 1406 | 508 | 1826 | 2274 | 6 | 3 | 8 | 9.75 |
| Time effect (b[se]) | 0.87 (1.07) | 0.001 (0.004) | **-0.02 (0.006)**\*\* | -0.01 (0.006) | 0.002 (0.002) | **0.07 (0.01)** \*\*\* | **-0.01 (0.002)**\*\*\* | -0.007 (0.004) | -0.05 (0.03) | 0.01 (0.01) |
| Percent change | - | - | -2.09% | - | - | 2.92% | -1.34% | - | - | - |

\*$p < .05$

\*\*$p < .01$

\*\*\*$p < .001$

EMAs on [their] phone or tablet" and 80.2% disagreed to some extent that the EMAs were "too long" each day. Over 80% strongly agrees that they gave "truthful information" on their surveys and disagreed to some extent that they were "uncomfortable answering questions

**Table 4. Participant study experience evaluation.**

| Question | Mean (SD) | Agree or Strongly Agree (%) |
|---|---|---|
| I enjoyed participating in this study | 4.38 (0.64) | 90.9 |
| Payment for my time during this study was fair | 4.25 (0.75) | 88.5 |
| It was easy for me to take surveys using my phone or tablet | 4.69 (0.53) | 98.6 |
| Each daily survey was too long | 1.83 (0.76) | 2.2 |
| I gave truthful information on my surveys | 4.82 (0.38) | 100.0 |
| I was able to communicate with study staff whenever I needed | 4.43 (0.66) | 90.5 |
| I felt supported by study team throughout the study | 4.45 (0.66) | 90.5 |
| I was uncomfortable answering questions about my urine and/or stool leaks | 1.72 (0.92) | 6.8 |
| I would participate in a study like this again in the future | 4.75 (0.46) | 87.7 |

about urine or stool leaks." Finally, three-quarters strongly agreed that they would participate in a study like this again in the future. Open-ended free text data support these themes; for example, one participant noted that they study was "quick and easy" and another liked that the study "did not require extensive involvement." Other participants suggested that receiving an SMS with a link to the survey was helpful for remaining "compliant."

Open-ended questions additionally revealed themes of gratitude over the focus on the topic of incontinence among this population. For example, one participant said that "more [studies like this] need to be conducted to understand difficulties of living with spina bifida and [dealing with] with bladder and bowel issues." Another respondent echoed this sentiment: "Not enough research like this is being done. . .[it feels like] doctors are 'throwing Jell-o at a tree'– with the tree being [those with SB]–to see what sticks in terms of helping." Two additional participants suggested, respectively, that they were grateful "for [the study's] taking the time to look into one our struggles that is least talked about," and that "there is not enough interest in adults with spina bifida." Five other participants emailed similar sentiments to the study manager after they had completed the study.

## Data capture

Commensurate with our presentation of the compliance data, we evaluated EMA's capabilities as a data collection tool by examining the volume of incontinence data at both the person- and EMA-levels.

At the *participant level*, 92.3% of individuals (84/91) reported any UI during the 30-day period and 79.1% (72/91) reported any FI during that same time frame. Focusing on the distribution of incontinence types, nearly all (97.1%: 89/91) were incontinent at some point during the study. Of these individuals, 19.1% (17/89) reported only UI, 5.6% (65/89) reported only FI, and 75.2% (67/89) reported both UI and FI. At the *EMA level*, any UI was reported on a total of 1163 days (45.1%: 1163/2578) and any FI was reported on a total of 508 days (19.7%: 508/2578). Focusing on the 1573/2578 days with any incontinence (61.0% of all days), 1065/1573 (67.7%) were UI only, 166/1573 (10.5%) were FI only, and 342/1573 (21.7%) were characterized with both UI and FI. Rates of opt-out for any of these questions were quite small (0.0%-0.8%).

## Data accuracy

We observed a high degree of accuracy between UI and FI reported in the EMA diaries and retrospective reports of any past week UI or FI in the exit surveys. The number of EMA days

on which UI was reported in the diaries was moderately correlated with reporting any past 4-week UI in the exit survey ($\rho$ = 0.690; $p < .001$), and the number of FI events reported in the EMA diaries was strongly correlated with exits survey reports of any FI in the past four weeks ($\rho$ = 0.813; $p < .001$) [61]. A majority (97.5%) of participants who reported any UI in the EMA diaries also reported past 4 week UI in the exit survey. Nearly all participants who noted any FI (98.5%) in the EMA diaries also reported FI in the exit survey.

### Suggestions for future work

Participants identified several areas on which the study could be improved in future iterations. These suggestions fell into three different areas: 1) conducting an additional, longer term study; 2) additional questions to contextualize incontinence; and 3) more questions about individual health conditions.

The need for a follow-up study was addressed via-a-vis the altering of "normal life" during COVID. For example, one participant said 'A lot of my normal activities living in [a major metropolitan city] were altered by COVID. Questions about social activities, including sex, would be answered differently pre-pandemic." Participants also though a longer time would capture a more representative picture of life with SB: "I have a few good weeks in between my bad weeks. So I'm more prone to accidents when my back and nerve pain flare up."

Participants provided suggestions for items to contextualize incontinence experiences in several ways. Multiple participants mentioned that the *volume* of food and/or drinks consumed often impacts both UI and FI. Two participants suggested adding questions about the degree of incontinence *urgency* as well as the *time of day* at which UI or FI happen. For example, one person noted "Sometimes I have feelings of urgency when I make it to the bathroom, and feelings of urgency when I don't make it to the bathroom. These situations impact my anxiety and stress differently [and also] how I choose to do or not dop activities." Another participant suggested asking about preparations in case of UI or FI: "[You should ask] Did you take the necessary items to change [if you went out]? Being prepared helps lower my stress."

Many suggestions about querying individual health conditions. One participant noted that their medications "impact [their] incontinence" while another said that aspects of her menstrual cycle, like bloating and pain, "put pressure on [their] bladder [making me] more nervous/stressed/anxious about leaks than [during] other times of the month, even if I didn't leak that day." Multiple participants wanted to see questions about daily physical symptoms: "General numbness, neuropathic numbness and pain levels could also be evaluated, as that can have effects on incontinence." Another participant said: "I know I feel more inclined to an accident when my legs feel like Jell-O."

### Discussion

Existing clinical and research data collection mechanisms typically lack the precision [12–15] to accurately describe the day-to-day experiences of UI and FI among individuals with SB. Because the success of interventions and/or education programs relies upon reliable and valid information, it is vital to engage data collection designs that capture UI and FI experiences with as much fidelity as possible to the ways in which individuals in this population actually live them. This preliminary study demonstrated that it is feasible to recruit an international sample of adults with SB to complete daily EMAs for a month that describe the behavioral and emotional context associated with their incontinence. Our results–scaffolded by suggestions from participants themselves–provide evidence of the likely success of longer term studies using this method.

An important pillar of a longitudinal study's success is the extent to which participants can be recruited and retained into a study, and the level to which their data completion is compliant with the proposed schedule. We engaged fully online recruitment methods to recruit international sample of participants from across the age spectrum and retained 97.8% of them through all three arms of data collection. In addition, participants were highly compliant with data submission expectations, completing 95% of the expected number of weekly-, participant- and study-levels. These levels exceed published recommendations of completion (80%) as a means of avoiding data bias [67] and are on par with or exceed completion levels in our past EMA studies [16, 17], as well as those estimated in recent meta-analyses of EMA studies (43%-95%) [68, 69]. We attribute several aspects of study design as helping to achieve this high compliance, including providing compensation for individual diary completion and for completion of a total of 90% of diaries. While it is possible that compensation inflated diary submission, it is also important to note that research staff provided participant support, such as in-depth EMA orientation and contacting participants who missed more than one EMA, also contributed to our compliance levels [16, 17].

Behavior reactivity can be a concern in EMA studies because outcomes of interest are repeatedly assessed close to the context in which they occur, potentially changing the types of data participants provide over the course of a study. Sensitive behaviors, like sexual activity, substance use, or in this case, incontinence, may be particularly vulnerable to reactivity [70, 71]. Our data did show small reporting differences (<2.5%) in the number of UI events and the number of daily activities reported as a function of day in the study. While no studies have assessed EMA reactivity in adults with SB, these findings are in line with reactivity volume observed in EMA studies of substance use [70], chronic conditions like tinnitus [72] diabetes [73] or pain [25] and our own work with sexual behavior [16, 17]. It will be important to continue to evaluate possible reactivity effects in this population as the study reporting period is longer.

Perhaps most importantly, participants highly rated their experience in the study. A primary intent of our person-centered study design was to facilitate participation for individuals whom may otherwise face substantial barriers to clinical research [32, 33]. All three study arms emphasized web-enabled technology that is commonly owned among adults with SB [38, 39]. Daily usage of these devices [40–43] meant that participants could participate in various aspects of data entry without much additional training on the data entry tasks associated with the study [44]. Moreover, the entirely electronic data collection structure allowed the cohort to flexibly provide their data "anywhere and anytime" in spaces that were convenient to them, thereby minimizing the potential disruption to their daily lives [36, 44]. We were able to capture an ample volume of incontinence events over the 30 days, and participants on average tended to strongly agree that the surveys were easy to take and not too long. These features–data entry ease and flexibility in how/when data entry happens–have been noted in other studies to support participant comfort in disclosing sensitive information [28], and participants in our study reported they felt comfortable answering questions about urine and/or stool leaks. While we did not assess whether there was any difference in comfort based on a specific study arm (e.g. enrollment survey vs. EMAs vs. exit survey), our data do provide evidence that engaging a user-centered design holds promise for ongoing research in this population.

Finally, it is important to consider how this framework could be applied to understand health experiences in populations with similar illness trajectories, such as individuals with other mild developmental delays (e.g. Down's syndrome), mobility issues (e.g. spinal cord injuries) or chronic incontinence associated with other health conditions (e.g. childbirth, cancer, Irritable Bowel Syndrome). EMA would be well suited, for example, to examine the severity of or life interference of chronic health conditions (e.g., pain [74]), or to assess the day-to-

day mental health impact of a condition [75]. Careful attention would need to be given to population and/or patient specific needs and limitations, such as ensuring the cognitive and physical ability to participate [76] or evaluating any need to strengthen basic digital literacy prior to participation (e.g. using/navigating a smart phone, using the internet, etc.) [77]. It would also be important to ensure–as in any study, but especially in those with long-term and/or chronic illness–that individuals' data are respected and kept private and secure, particularly when discovery could adversely impact other aspects of daily life [78].

## Limitations

As with any study, there are limitations to consider. This study was limited to individuals 18 years and older. Incontinence can be a lifetime challenge for those with SB, and the nature of how it is experienced in adulthood may have links to experiences in earlier life stages. Future studies may benefit from inclusion of younger individuals as a means of situating incontinence within a developmental life stage lens. Moreover, our sample was primarily White, heterosexual, well-educated, and exhibited a high degree of living and mobility independence. There is possibility of self-selection bias within this set of characteristics, meaning that individuals who are more caregiver-dependent, or those with greater degrees of developmental delay, may have been less likely to be able to participate in the study. However, our participants had reasonably similar gender and neurological characteristics to other samples from SB-focused research [7, 79–81]. It will be important in future studies to understand the extent to which an EMA framework would need to be adapted to accommodate individuals with SB from different cultural or socioeconomic backgrounds. For example, participants in this study used their own phone for the duration of the study. Individuals experiencing financial hardship may not own a phone or have internet. One solution could be to build the cost of a temporary phone and monthly wireless service into the study design [82].

There are also limitations associated with the study design. Because our aims were to evaluate method feasibility, we did not assess the extent to which differences in design could impact the data collected. It will be important for future research to evaluate how cadence of EMA completion (for example, once per day vs. more frequently), a longer study length (e.g. 30 days vs. 120 days) or a different data collection schedule (e.g. continuous vs. weeks on/weeks off). Many of our participants noted that a longer study length would better capture the variability of their "daily lives," but it remains unclear both how many more weeks of collection and how frequently these assessments would need to occur to fully gauge the bandwidth of experiences. We also engaged a combination of several different convenience and snowball sampling strategies to reaching potential participants. In addition to the usual caveats about non-probability sampling approaches, it is possible that some participants recruited from the same networks (e.g., members of the same SB state chapter, or patients in the same clinic) may have shared characteristics that further limited broad generalizability of our findings to individuals with SB who are outside of those networks.

Finally, it is impossible to verify if any participants provided answers with someone else's help, and/or to what extent such help may have influenced answered given. Questions about both the context of incontinence itself, as well as how people feel about it, can be deeply personal and raise embarrassment. While individuals can be generally asked about incontinence in clinical encounters, we assessed specific dimensions about urine and stool leaks that could have made some people uncomfortable. However, participants were encouraged during the enrollment process to provide data in private settings, and the survey was built to allow individuals to opt out of most questions. Rates of opt-out were quite small (0.0%-0.8%), suggesting

that most people who took the survey had little trouble with the content and nature of the questions.

## Conclusion

The ability to keep participants engaged in research over an extended period of time is vital to capturing both a sufficient volume of data around outcomes of interest and the quality of data that reliably and validly reflect the daily experiences of the participants themselves. This study provided evidence that it is feasible to use electronic approaches to recruit a cohort of adults with SB to participate in a study to report daily incontinence via web-enabled data collection. Our cohort was highly compliant with study protocols, highly rated their study experience and provided suggestions on how to improve study approaches moving forward. We suggest that these data provide great promise for ongoing research in a population that can face barriers to clinical research participation.

## Acknowledgments

We are grateful to the participants of this study for providing their information, and to Pat Brooks for outstanding support of our participants during the study.

## Author Contributions

**Conceptualization:** Devon J. Hensel.

**Data curation:** Devon J. Hensel.

**Formal analysis:** Devon J. Hensel, Audrey I. Young, Konrad M. Szymanski.

**Funding acquisition:** Devon J. Hensel, Konrad M. Szymanski.

**Investigation:** Devon J. Hensel, Konrad M. Szymanski.

**Methodology:** Devon J. Hensel.

**Project administration:** Devon J. Hensel, Konrad M. Szymanski.

**Resources:** Audrey I. Young.

**Software:** Devon J. Hensel, Konrad M. Szymanski.

**Validation:** Konrad M. Szymanski.

**Writing – original draft:** Devon J. Hensel, Konrad M. Szymanski.

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
