## [Decision Letter · Decision Letter 0]

24 Jul 2023

PONE-D-23-11493The Feasibility of Using Ecological Momentary Assessment to Understand Urinary and Fecal Incontinence Experiences in Adults with Spina Bifida: A 30-day Diary StudyPLOS ONE

Dear Dr. Hensel,

Thank you for submitting your manuscript to PLOS ONE. After careful consideration, we feel that it has merit but does not fully meet PLOS ONE’s publication criteria as it currently stands. Therefore, we invite you to submit a revised version of the manuscript that addresses the points raised during the review process.

 Two reviewers have assessed the manuscript, and are generally positive about the study. Below, you can find their comments on revisions that would be helpful to do in order to improve the presentation of the study.

We look forward to receiving your revised manuscript.

Kind regards,

Hanna Landenmark

Staff Editor

PLOS ONE

Journal Requirements:

“Dr. Hensel is a paid research consultant with For Goodness Sake, LLC.”

5. Please include your tables as part of your main manuscript and remove the individual files. Please note that supplementary tables (should remain/ be uploaded) as separate "supporting information" files.

Reviewers' comments:

Reviewer's Responses to Questions

**Comments to the Author**

1. Is the manuscript technically sound, and do the data support the conclusions?

Reviewer #1: Yes

Reviewer #2: Yes

2. Has the statistical analysis been performed appropriately and rigorously? 

Reviewer #1: Yes

Reviewer #2: Yes

3. Have the authors made all data underlying the findings in their manuscript fully available?

Reviewer #1: No

Reviewer #2: Yes

4. Is the manuscript presented in an intelligible fashion and written in standard English?

Reviewer #1: Yes

Reviewer #2: Yes

5. Review Comments to the Author

Reviewer #1: Excellent paper. However, the authors did not display the 4 tables which were referred in the results section. I also went to the OSF website to look at the data available, but could not find the data summarized in the table format.

Considering that the sample was an international one and English-speaking, I suggest in the Discussion section, that the authors incorporate some evidence in terms of differences and similarities between SB populations having different cultural, ethnic and socioeconomic backgrounds. If possible, it would be nice to bring some research implications when using electronic approaches to engage SB population which maybe have "similar" illness trajectory as well as needs, but could also be impacted differently depending on their digital literacy, and cultural perception about their incontinence(s).

Reviewer #2: In general:

The authors present a secondary analysis of a larger prospective study to evaluate the feasibility of using ecological momentary assessment to understand urinary and fecal incontinence in adults with spina bifida. This study is very interesting and relevant to this area of research. The presented results will have an impact on the clinicians taking care of adults with spina bifida. This reviewer would like to suggest the following for consideration.

In detail:

Line 29:

Are the provided keywords MeSH terms?

Line 35:

Information is missing here. Please provide name of agency and grant number as highlighted in lines 606 – 609.

Line 172:

Please provide information regarding the registration of the ‘larger prospective study’

Line 324:

Regarding “Raw data files are stored with the Open Science Framework [65].”, will the authors provide a link in the final version, so that anyone can access the data?

Regarding all tables:

The authors present mean with SD in table 1, 2, and 4 but median in table 3. Did the authors check for normal distribution? If so, please provide information on that! Thank you.

Table 1:

Regarding “Hispanic/Latino (yes)”, what is the reasoning to dichotomize the cohort in this way, given that the authors already provided detailed information with respect to race?

Table 2:

- Please provide units for “Time to complete diaries”.

- Regarding “Gender Identity”, why using the term gender here, while using birth-assigned sex in table 1?

- Regarding “Sexual Orientation”, isn’t that quite an unfortunate way to dichotomize a cohort in either “Heterosexual” or “Sexual minority”?

6. PLOS authors have the option to publish the peer review history of their article (what does this mean?). If published, this will include your full peer review and any attached files.

Reviewer #1: No

Reviewer #2: **Yes: **Matthias Walter

---

## [Author Response · Author response to Decision Letter 0]

27 Jul 2023

Reviewer #1: 

Excellent paper. 

However, the authors did not display the 4 tables which were referred in the results section. I also went to the OSF website to look at the data available, but could not find the data summarized in the table format.

We apologize for this oversight and have included the table for this revision. 

2. Considering that the sample was an international one and English-speaking, I suggest in the Discussion section, that the authors incorporate some evidence in terms of differences and similarities between SB populations having different cultural, ethnic and socioeconomic backgrounds. 

There is very little large data available with which we could make this comparison. We do state in the limitations that: “Moreover, our sample was primarily White, heterosexual, well-educated, and exhibited a high degree of living and mobility independence. There is possibility of self-selection bias within this set of characteristics, meaning that individuals who are more caregiver-dependent, or those with greater degrees of developmental delay, may have been less likely to be able to participate in the study. However, our participants had reasonably similar gender and neurological characteristics to other samples from SB-focused research.”

To that sentence we have now added: “It will be important in future studies to understand the extent to which an EMA framework would need to be adapted to accommodate individuals with SB from different cultural or socioeconomic backgrounds. For example, participants in this study used their own phone for the duration of the study. Individuals experiencing financial hardship may not own a phone or have internet. One solution could be to build the cost of a temporary phone and monthly wireless service into the study design. 

3. If possible, it would be nice to bring some research implications when using electronic approaches to engage SB population which maybe have "similar" illness trajectory as well as needs, but could also be impacted differently depending on their digital literacy, and cultural perception about their incontinence(s).

We agree and thank the reviewer for this suggestion. We have added text in the Discussion to read: “Finally, it is important to consider how this framework could be applied to understand health experiences in populations with similar illness trajectories, such as individuals with other mild developmental delays (e.g. Down’s syndrome), mobility issues (e.g. spinal cord injuries) or chronic incontinence associated with other health conditions (e.g. childbirth, cancer, Irritable Bowel Syndrome). EMA would be well suited, for example, to examine the severity of or life interference of chronic health conditions (e.g., pain [74]), or to assess the day-to-day mental health impact of a condition [75]. Careful attention would need to be given to population and/or patient specific needs and limitations, such as ensuring the cognitive and physical ability to participate [76] or evaluating any need to strengthen basic digital literacy prior to participation (e.g. using/navigating a smart phone, using the internet, etc.) [77]. It would also be important to ensure – as in any study, but especially in those with long-term and/or chronic illness – that individuals’ data are respected and kept private and secure, particularly when discovery could adversely impact other aspects of daily life [78].”

Reviewer #2: 

In general:

The authors present a secondary analysis of a larger prospective study to evaluate the feasibility of using ecological momentary assessment to understand urinary and fecal incontinence in adults with spina bifida. This study is very interesting and relevant to this area of research. The presented results will have an impact on the clinicians taking care of adults with spina bifida. This reviewer would like to suggest the following for consideration.

1. In detail:

 Line 29:

 Are the provided keywords MeSH terms?

We thank the reviewer for their attention to this point - they are not, and we have now changed the keywords that were not in the MeSH database. 

2. Line 35:

 Information is missing here. Please provide name of agency and grant number as highlighted in lines 606 – 609.

We apologize for this oversight and have added the information (lines 35-38).

3. Line 172:

 Please provide information regarding the registration of the ‘larger prospective study’

The study was not registered as it was not a clinical trial. 

4. Line 324:

 Regarding “Raw data files are stored with the Open Science Framework [65].”, will the authors provide a link in the final version, so that anyone can access the data?

We have now provided a link - on the title page and in the references link. We thank the reviewer for their attention to this point.

5. Regarding all tables:

 The authors present mean with SD in table 1, 2, and 4 but median in table 3. Did the authors check for normal distribution? If so, please provide information on that! Thank you.

We thank the reviewer for this question. For age (the only continuous variable in Table 1) we now include the Mean (SD) and median (IQR). 

We now include the text (lines 364 - 367): A preliminary examination of the distribution of completion time at the weekly level (number of entries: Mean: 7.2; Median: 7.0; Mode: 7.0) and the participant level (Mean: 29.3; Median: 30.0; Mode: 30.0) suggested that the variable was normally distributed. Therefore, values in Table 2 and 3 are represented by the mean. We also include (completion time: lines 380 - 383): A preliminary examination of the distribution at the and weekly level (Mean: 35.4; Median: 2.0; Mode: 2.0) and participant level (Mean: 36.1; Median: 2.0; Mode: 2.0) suggested that the variable was positively skewed. Therefore, values in Table 2 and 3 are represented by the median. 

6. Table 1:

 Regarding “Hispanic/Latino (yes)”, what is the reasoning to dichotomize the cohort in this way, given that the authors already provided detailed information with respect to race?

Both race and ethnicity were collected as required for federally funded studies - items are presented in Table 1 as they were in the collection format required by NIH. 

7. Table 2:

 - Please provide units for “Time to complete diaries”.

The units are minutes, and we now include this in Table 2. We thank the reviewer for pointing this out. 

8. Regarding “Gender Identity”, why using the term gender here, while using birth-assigned sex in table 1?

Birth assigned sex generally refers to biology - being male or female based on genitals and chromosomes. Gender identity generally refers to how someone feels on the inside about their gender and how they socially express those feelings (e.g., clothing, mannerisms, behaviors). Sometimes gender identity matches birth assigned sex, and sometimes it does not. We use these terms to acknowledge the complexity of how people think about themselves. While gender is not specifically a question of interest in this paper, we include it to show the demographic profile of participants - much like we include other aspects of sexuality such as orientation. 

9. Regarding “Sexual Orientation”, isn’t that quite an unfortunate way to dichotomize a cohort in either “Heterosexual” or “Sexual minority”?

We realized in pondering this question that we had neglected to include sexual orientation in Table 1, which we have now done. As displayed, nearly all (88.0%) of participants reported themselves as heterosexual. Because of this large majority, we do not have the power to analyze the sexual minority group by specific self-identified categories. We hope this now clears up the issue.

---

## [Decision Letter · Decision Letter 1]

27 Sep 2023

The Feasibility of Using Ecological Momentary Assessment to Understand Urinary and Fecal Incontinence Experiences in Adults with Spina Bifida: A 30-day Study

PONE-D-23-11493R1

Dear Dr. Hensel,

We’re pleased to inform you that your manuscript has been judged scientifically suitable for publication and will be formally accepted for publication once it meets all outstanding technical requirements.

Kind regards,

Sandip Varkey George, PhD

Academic Editor

PLOS ONE

Additional Editor Comments (optional):

Reviewers' comments:

Reviewer's Responses to Questions

**Comments to the Author**

1. If the authors have adequately addressed your comments raised in a previous round of review and you feel that this manuscript is now acceptable for publication, you may indicate that here to bypass the “Comments to the Author” section, enter your conflict of interest statement in the “Confidential to Editor” section, and submit your "Accept" recommendation.

Reviewer #2: All comments have been addressed

2. Is the manuscript technically sound, and do the data support the conclusions?

Reviewer #2: Yes

3. Has the statistical analysis been performed appropriately and rigorously? 

Reviewer #2: Yes

4. Have the authors made all data underlying the findings in their manuscript fully available?

Reviewer #2: Yes

5. Is the manuscript presented in an intelligible fashion and written in standard English?

Reviewer #2: Yes

6. Review Comments to the Author

Reviewer #2: Thank you for addressing all comments sufficiently. This reviewer does not any further requests / comments.

7. PLOS authors have the option to publish the peer review history of their article (what does this mean?). If published, this will include your full peer review and any attached files.

Reviewer #2: **Yes: **Matthias Walter

---

## [Editor Report · Acceptance letter]

10 Oct 2023

PONE-D-23-11493R1 

The Feasibility of Using Ecological Momentary Assessment to Understand Urinary and Fecal Incontinence Experiences in Adults with Spina Bifida: A 30-day Study 

Dear Dr. Hensel:

I'm pleased to inform you that your manuscript has been deemed suitable for publication in PLOS ONE. Congratulations! Your manuscript is now with our production department. 

Kind regards, 

on behalf of

Dr. Sandip Varkey George 

Academic Editor

PLOS ONE